# Comparative Analysis of CT Texture in Lumbar and Femur and Its Correlation with Bone Mineral Density and Content over Time: An Exploratory Study

**DOI:** 10.3390/diagnostics13233588

**Published:** 2023-12-03

**Authors:** Min Woo Kim, Young Min Noh, Jung Wook Huh, Han Eol Seo, Dong Ha Lee

**Affiliations:** Department of Orthopedic Surgery, Busan Medical Center, Busan 47527, Republic of Korea; drkimminwoo@naver.com (M.W.K.); doctornoh77@naver.com (Y.M.N.); gizer00@hanmail.net (J.W.H.); mdseo86@gmail.com (H.E.S.)

**Keywords:** dual-energy X-ray absorptiometry (DXA), Computed Tomography Hounsfield Unit (CT HU), Bone Mineral Content (BMC), morphometric texture analysis, linear regression

## Abstract

Background: This research explores the application of morphometric texture analysis in chest Computed Tomography (CT) scans for determining Bone Mineral Content (BMC) and its temporal changes, both crucial in diagnosing osteoporosis. The study establishes an innovative approach to osteoporosis screening by leveraging Hounsfield Units (HUs) in CT scans to evaluate BMC, offering a comparison with dual-energy X-ray absorptiometry (DXA)-based BMC. Methods: A total of 806 instances (encompassing 379 individuals) were meticulously compiled from a sole institution, during the period stretching from 6 May 2012 to 30 June 2020. In this detailed analysis, each participant was subjected to a pair of chest CT scans, sequentially pursued by a DXA scan, spread over two years. Focused records of BMC values at the inaugural lumbar vertebra (L1) were secured from both the DXA and CT axial slices across all instances. A meticulous selection process pinpointed the largest trabecular section from the L1 vertebral body, whereupon 45 distinctive texture attributes were harvested utilizing gray-level co-occurrence matrix methodologies. Utilizing these amassed 45 attributes, a regression architecture was devised, aiming to forecast the precise BMC values individually. Moreover, an alternative regression framework was engaged, leveraging 90 distinct features, to gauge the BMC fluctuations observed between the duo of scans administered to each participant. Results: The precision of the cultivated regression frameworks was scrupulously assessed, benchmarking against the correlation coefficient (CC) and the mean absolute deviation (MAE) in comparison to the DXA-established references. The regression apparatus employed for estimating BMC unveiled a CC of 0.754 and an MAE of 1.641 (g), respectively. Conversely, the regression mechanism devoted to discerning the variations in BMC manifested a CC of 0.680, coupled with an MAE of 0.528 (g), respectively. Conclusion: The innovative methodology utilizing morphometric texture analysis in CT HUs offers an indirect, yet promising, approach for osteoporosis screening by providing estimations of BMC and its temporal changes. The estimations demonstrate moderate positive correlations with DXA measures, suggesting a potential alternative in circumstances where DXA scanning is limited.

## 1. Introduction

Osteoporosis, a prevalent bone disease, is characterized by a decrease in bone mass and strength, leading to an increased risk of fractures—particularly in trabecular bone-rich areas such as the proximal femur, vertebral body of the spine, and the distal radius [1]. This necessitates the need for a comprehensive diagnostic system that can predict and manage osteoporosis effectively.

Currently, dual-energy X-ray absorptiometry (DXA) is the primary tool for evaluating Bone Mineral Density (BMD) and Bone Mineral Content (BMC). However, the assessment of BMD and BMC can be complicated due to their variability, which depends on factors such as the progression of osteosclerosis and the degree of adiposity [2]. The lumbar spine, in particular, poses significant challenges to accurately measure BMD and BMC when compared to the femoral neck bone density [3].

In this innovative study, we utilize axial cuts from chest CT scans at the L1 level and from abdomen pelvis CT scans at the femur neck. The aim is to gain a comprehensive understanding of bone health by examining the correlation between BMD, BMC, and texture analysis values derived from these CT scans.

BMC plays a vital role in fracture prediction due to its influence on bone strength, structure, and fragility, whereas BMD is a key determinant of bone density and risk of osteoporosis [4]. The trabecular bone, due to its high bone turnover rate, is especially sensitive to metabolic stimuli [5]. Therefore, BMC and BMD hold significant potential as precise indicators for osteoporosis and fracture risk, particularly when considering the trabecular bone.

Our novel approach exploits the potential of CT scans that fall outside DXA insurance guidelines, unveiling a more efficient method for evaluating bone mineral status. This method is especially beneficial for patients who do not meet the criteria for DXA. Our comprehensive data set, which includes follow-up patients, offers unique insights into the progression of bone health over time. We further quantify the interval changes in BMD and BMC and examine their correlation with texture analysis values.

This study aims to establish an objective basis for quantifying the degree of BMD and BMC at different scan intervals and tracking their temporal changes using CT models. The methodology involves an in-depth analysis of HU values in CT, the extraction of model-based texture features via a Gray-level Co-occurrence Matrix (GLCM), and the formulation of Linear Regression models informed by DXA measures [6,7]. Our findings underscore the considerable potential of texture analysis as a reliable and efficient tool for monitoring bone mineral status, ultimately leading to more precise osteoporosis diagnosis and management.

## 2. Materials and Methods

### 2.1. Subjects for the Region of Interest

Our research, sanctioned by the institutional review board (P01-202109-21-014), began with an extensive compilation of 3620 instances, incorporating 1643 individuals. Every participant underwent dual CT scans—encompassing both chest and abdomen-pelvis sectors—and DXA, all within the same facility, from 9 May 2011 to 30 July 2022. From this extensive assortment, a rigorous selection regimen was executed, yielding a refined assemblage of 892 instances involving 379 participants, conforming to stringent prerequisites: a chronological divergence under a month between CT and DXA evaluations, a minimum of two subsequent follow-ups, and a two-year lapse post-initial evaluation.

As the selection evolution progressed, instances were omitted contingent on the fulfillment of particular criteria: the non-presence of a tangible, quantifiable axial section of the L1 vertebra or femur neck within CT visuals; antecedent records of L1 compression; pronounced fractures or intrusive surgical reparations due to fracturing events; the discernment of metallic discrepancies resulting from precarious burst fractures; or complexities in isolating trabecular bones owing to pronounced osteolytic alterations or pathological manifestations.

The culmination of this exclusionary phase saw the subtraction of 350 instances (embodying 211 individuals), ushering in the conclusive selection of 806 instances from 379 participants for in-depth analysis (Figure 1). This meticulously curated selection procedure bolstered the integrity and dependability of our investigative endeavor, centering on the paramount, elucidative instances for our exploratory journey into the applicative frontier of texture scrutiny in the surveillance of bone mineral integrity.

### 2.2. CT and DXA Imaging Protocols

The CT scans were meticulously performed utilizing a Siemens scanner (SOMATOM 128, Definition AS+; Siemens Healthcare, Forchheim, Germany), adhering faithfully to a predefined standard protocol. Each scanning procedure was executed as a single-energy CT scan, with operational parameters meticulously calibrated to 120 kVp and 247 mA, incorporating a dose modulation with a 0.6 mm collimation. An effective pitch was firmly established at 0.8, paired with the deployment of a B60 (sharp) reconstruction kernel. Specifically, for the chest CT scans conducted in the absence of contrast, a consistent reconstructed slice thickness of 5.0 mm was diligently upheld.

In the case of the DXA scans, a conventional apparatus was employed, executing scans in strict adherence to a traditional protocol (GE Lunar Prodigy, GE Healthcare, Wauwatosa, WI, USA). Subsequent reports were synthesized utilizing specialized vendor-affiliated software (Physicians Report Writer DX; Hologic, Marlborough, MA, USA). Such rigorous conformity to standardized imaging protocols underscores the replicability and unwavering consistency integral to the outcomes of our investigative endeavor.

### 2.3. Regions of Interest

The specific areas targeted for statistical analysis within the bone images were meticulously limited to the trabecular sections of the bone, a strategic move to curtail any potential distortions in the ensuing measurements. Faced with multiple methodologies for the demarcation of these Regions of Interest (ROIs), our choice gravitated towards employing the thresholding technique for the pursuits of this study. For each individual involved in the study, a singular two-dimensional (2D) slice image was conscientiously chosen from the CT axial cross-sections. These selected images predominantly featured the most expansive axial trabecular regions either of the L1 spinal body or the femoral neck. Figure 2 illustratively encapsulates our process, showcasing the texture analysis unfolding primarily within a circular perimeter, which encompasses the vast majority of the trabecular space under consideration.

### 2.4. Feature Extraction

We meticulously extracted a compilation of 45 distinct features from the designated Regions of Interest (ROIs), which encompassed five pivotal intensity-oriented attributes, harvested via histogram analysis, in conjunction with 40 texture-centric features, cultivated from a GLCM (Gray-level Co-occurrence Matrix) foundation [8,9]. These meticulously garnered features were subsequently integrated into two predominant modeling paradigms: a Linear Regression (LR) construct and an Artificial Neural Network (ANN) framework. The LR construct was adept at prognosticating BMC and BMD, leveraging a linear amalgamation of the 45 infused input parameters. Concurrently, the ANN model was architecturally structured as a comprehensively interconnected neural network, comprising hierarchical layers where the inaugural trio of hidden layers were characterized by configurations of eight, eight, and two nodes, each synergistically interacting through a non-linear rectified linear unit operator, as visually elucidated in Figure 2.

The intensity-focused attributes were ingeniously captured, utilizing the histogram of the ROI imagery, encapsulating essential metrics such as mean, standard deviation, skewness, kurtosis, and entropy. These cardinal parameters resonated with intrinsic bone intensity characteristics, like luminance, asymmetry, randomness, homogeneity, and acuteness. Augmenting this, a suite of 40 textured attributes was unveiled through intricate texture examinations, designed to unveil nuanced spatial interrelationships between contiguous pixel entities within a 2D visualization. Originating from the foundational GLCM as illustrated in Figure 2, and predicated on a spectrum of *n* grayscale gradients and a horizontal orientation, the resultant matrix manifested dimensions of *n* × *n*. Each constituent element within this matrix, demarcated as the (i,j)th element, echoed the cumulative occurrences of horizontally juxtaposed pixels, registering grayscale values of i and j within a discretely normalized ROI imagery realm, characterized by an intensity gradient fluctuating between 1 and *n*. In our exploratory odyssey, an ensemble of eight multifaceted GLCMs was meticulously crafted for each illustrative ROI instance, navigating across four distinctive gradients (*n* = 16, 32, 64, 128) and bifurcating directional orientations (horizontal and vertical), while encapsulating five intrinsic statistical metrics: entropy, contrast, correlation, homogeneity, and variance. Each metric resonates with individual GLCMs.

### 2.5. Estimation of BMD and BMC Change Using CT

Figure 2 visualizes our BMC estimation approach. From each CT scan, we picked one slice that showed the largest view of the trabecular bone area. In each chosen slice, 45 different features were extracted. Five of these were based on intensity histograms, while the remaining 40 utilized the Gray-level Co-occurrence Matrix (GLCM), a common technique in texture analysis.

GLCM, as explained in Table 1, helps in understanding an image’s texture by considering the frequency of specific pixel pair values [8,9]. We used a combination of multiple statistics in both histogram and GLCM, applying a specific formula to assign each a unique feature index, _j. We ended up with 90 features per patient from two separate chest CT scans. Using MATLAB, various mathematical operations were applied. Two linear regressors were developed. The first regressor used the 45 features to estimate BMC and BMD values, and the second used 90 features to monitor changes over time. The calculations involved a mathematical formula that combined these features linearly.
diagnostics-13-03588-t001_Table 1Table 1Gray-level Co-occurrence Matrix feature parameters.Analytical ToolParameterValue/Name/FunctionFeature #HistogramStatistics (k)mean (k = 1), standard deviation (k = 2), skewness (k = 3), kurtosis (k = 4), entropy (k = 5)5Texture (GLCM)Directions (l)horizontal (l = 1), vertical (l = 2)2 × 4 × 5 = 40Levels (m)16 (m = 1), 32 (m = 2), 64 (m = 3), 128 (m = 4)Statistics (*n*)contrast (*n* = 1), correlation (*n* = 2), energy (*n* = 3), homogeneity (*n* = 4), variance (*n* = 5)
y_hat_j = sum (from j = 1 to J) of w_j ∗ x_j + b

To improve the model’s clarity and usability, the Least Absolute Shrinkage and Selection Operator (LASSO) method was utilized, incorporating a penalty term represented by _*λ* in the model. The use of LASSO, governed by the equation below, allows the model to be simplified based on the value of _*λ*.
{w_j*, b*} = argmin [sum (from i = 1 to I) of (y_i − y_hat_i)^2 + lambda ∗ sum (from i = 1 to I) of absolute value of x_i]

### 2.6. Correlation Assessment

We utilized Linear Regression (LR) and a comprehensive Artificial Neural Network (ANN) to predict Bone Mineral Content (BMC) and Bone Mineral Density (BMD), and to analyze the correlation between the predicted values and the actual DXA BMD values. Let us consider x_ij as the i-th feature value for the j-th sample (case) and y_j as the BMD reference for the j-th sample. During preprocessing, each sample was normalized: x_ij_new = (x_ij − mean(x_i))/stddev(x_i), where mean(x_i) and stddev(x_i) are the average and standard deviation of the i-th feature, respectively. Similarly, each reference was normalized as y_j_new = (y_j − mean(y))/stddev(y), where mean(y) and stddev(y) are the overall mean and standard deviation, respectively.

In the LR model, the BMD was estimated as a weighted sum of 45 features plus a bias: y_pred_j = w_0 + Σ(w_i ∗ x_ij_new). Each weight, w_i, was determined by minimizing the Mean Squared Error (MSE): MSE = Σ(y_j_new − y_pred_j)^2. This was performed using a pseudoinverse in a normal equation form. Since there were more samples than trainable parameters, overfitting was not a concern; therefore, there was no need to split the data into training and test sets or to use regularization techniques such as ridge regression or LASSO.

## 3. Results

### Patient Demographics

This study encompassed an analysis of 806 samples from 379 participants, with 175 males and 204 females included. The average age and BMI of the subjects were 54.52 ± 7.56 years and 23.29 ± 5.65 kg/m^2^, respectively. Time intervals averaged 0.89 ± 5.22 days between chest CT and DXA scans, and 1.92 ± 6.89 days between APCT and DXA scans. The average durations between the initial and final chest CT scans were 1412 ± 44.57 days, and between the first and last DXA scans, it was 1348 ± 31.68 days, detailed in Table 2.

Figure 3 and Figure 4 display the relationship between estimated and actual DXA measurements of BMC and BMD at the L1 spine and hip, showcasing the correlation coefficients and MSE values. The coefficients (MSE) from the LR model fluctuated between 0.760 (1.397) and 0.897 (0.856). A substantial correlation was discovered, not just in the absolute BMC and BMD values, but also in their temporal alterations, with a Pearson correlation coefficient (*r*) of 0.918 for BMC and BMD, and 0.654 for their temporal changes, as elucidated in Figure 5.

As per Section 2.3, estimates were exclusively obtained from CT images through feature extraction and Linear Regression methodologies, as displayed in Figure 6. Here, the correlation coefficient and mean absolute error for BMC change regression were 0.654 and 0.528 (g), respectively, and the paired *t*-tests gave *p*-values of 1, confirming the unbiased nature of the LR model. Figure 6 also indicates that a higher l_1 penalty resulted in a lower correlation coefficient and a higher mean absolute error, driving many weights to zero. The BMC regressor exhibited a minor performance decrement due to the penalty, attaining a correlation coefficient of about 0.65 with six features at a λ of 0.04, whereas a more pronounced performance loss was seen in the BMC change regressor, emphasizing the distinct contribution of each feature.

## 4. Discussion

In a stride towards optimizing the precision and efficacy of osteoporosis detection, our approach integrated machine learning techniques along with a straightforward LR model applied to the texture analysis of CT HUs. Our endeavor led to the successful formulation of LR models proficient in estimating the BMC and monitoring its temporal variations, signifying a noteworthy progression in medical imaging domains.

To validate our outcomes, emphasis was placed on the robust correlation observed between estimates derived from CT HUs and authentic measurements obtained via DXA. Figure 3 and Figure 4 lucidly present this association, showcasing the BMC and BMD estimates procured from the L1 axial slices and the hip regions, respectively. A remarkable observation was the superior correlation achieved when the estimations from these ROIs coincided with the respective DXA measurement locales, manifesting correlation coefficients between 0.760 and 0.897, and MSE values, validating the competency of our LR model in BMC and BMD prediction.

Our research not only elucidated a substantial correlation between the absolute BMC and BMD values, with a correlation coefficient (*r*) of 0.918, but also unveiled a meaningful association concerning their temporal changes, bearing a correlation coefficient of 0.654. This insight is immensely valuable for it facilitates the continuous surveillance of osteoporosis evolution, a crucial aspect for the proficient administration and therapeutic strategy concerning the ailment.

Expanding upon these findings, our study unveils promising directions for incorporating machine learning into medical imaging. Utilizing conventional radiomics procedures, encompassing pre-processing, manual segmentation, and feature extraction, we achieved successful BMD predictions, indicating that CT HU texture analysis could emerge as a potent tool for BMC estimations, presenting a viable alternative to traditional DXA imaging methodologies.

In our research, we leveraged machine learning’s capabilities, particularly focusing on Artificial Neural Network (ANN) and a direct Linear Regression (LR) model, aiming to augment the accuracy and efficiency of our analysis. The advent of substantial advancements in computational capabilities has positioned machine learning as a revolutionary influence across various sectors, including its remarkable impact on enhancing diagnostic precision in medical imaging [10,11].

Radiomics, a machine learning subset, has experienced substantial growth, attributed to its ability to quantifiably extract features from designated Regions of Interest (ROIs) within images, which are instrumental in accomplishing predictive or prognostic goals. Our study employed fundamental radiomics procedures, including pre-processing, manual segmentation, and feature extraction, focusing primarily on predicting Bone Mineral Density (BMD). The features emphasized in our study comprised energy, kurtosis, and skewness from intensity, alongside texture analysis utilizing the Gray-level Co-occurrence Matrix (GLCM) [12].

GLCM has garnered extensive acceptance due to its proficiency in extracting diverse tissue features by evaluating the occurrence frequency of pixel pairs in specified relationships. The functions extracted from GLCM encapsulate a spectrum of texture attributes such as energy, contrast, entropy, autocorrelation, correlation, inverse moment, and cluster shade [13]. The convergence of machine learning with radiomics has witnessed recent endorsements, particularly prevalent in Magnetic Resonance Imaging (MRI) studies [14]. For example, GLCM textures and logistic regression have been instrumental for researchers in distinguishing specific brain tumor varieties. Numerous studies have diversified their feature assortment utilizing approaches such as Gray-level Run Length Matrices (GLRLM) and wavelet transformations [15]. Employing lower-dimensional handcrafted features extracted from high-dimensional images as inputs for deep learning models facilitates the model’s structural simplification, mitigating overfitting risks when image samples are limited [16].

While our study illuminates the promise of CT HU texture analysis in forecasting BMC and tracking its temporal variations, acknowledging certain limitations is essential for a nuanced understanding. First and foremost, our research did not incorporate specific osteoporosis risk factors or the impact of certain medications that might expedite changes in BMC. Such omissions could influence the ultimate precision and relevance of our findings, as these variables can considerably affect BMC values and their fluctuations.

Secondly, our Region of Interest (ROI) model predominantly centered on the L1 axial cut obtained from chest CT scans. Even though the L1 axial cut is a customary component in DXA scans for Bone Mineral Density (BMD) evaluations, our exclusive dependency on this singular cut may curtail the all-encompassing nature of our conclusions. Our APCT studies also specifically zeroed in on the femur neck, overlooking additional regions that might have unveiled further valuable insights.

Moreover, in juxtaposing our outcomes with those from DXA measurements, our focus remained narrowly confined to the L1 value, bypassing the conventionally assessed L1–L4 range intrinsic to DXA investigations. Also, the absence of a radiologist in our team may have limited the scope of our radiological analysis. Future research could be enriched by the inclusion of imaging experts to broaden the interpretive perspectives. Finally, a possible inclination towards bias might permeate our study, attributed to the predominant reliance on data sourced from a solitary institution. Such a factor could potentially circumscribe the broader applicability and extrapolation of our results to diverse settings or demographic groups. Compounding this, the relatively abbreviated follow-up durations featured in our study might not be illustrative of protracted BMC alterations. A more extended observation of these shifts could unveil insights of heightened pertinence for the meticulous identification and continual observation of osteoporosis. Despite these limitations, our study provides an innovative approach to BMC estimation and can guide future research towards refining and expanding this methodology. Further studies involving larger, more diverse datasets and longer follow-up periods could potentially validate and enhance the effectiveness of CT HU texture analysis in osteoporosis detection and monitoring.

In summary, our research successfully employs the texture analysis of CT Hounsfield Units (HUs) to establish linear regression models for estimating Bone Mineral Content (BMC) and monitoring its changes. Among the various factors analyzed, the 45 CT texture analysis features stand out as a key single feature that plays a crucial role in effectively representing the BMC equivalent value and its changes. These features are integral to the model’s ability to mimic the BMC values and track their variations, thereby contributing significantly to the model’s robustness and accuracy. The high correlation between these estimates and actual DXA values testifies to the efficacy and precision of our approach. Moreover, the relatively short scan interval between the CT and DXA, compared to previous studies, further reinforces the reliability of our results. These findings lay the groundwork for utilizing CT scans alone in predicting BMC, providing a fresh and efficient perspective in the field of osteoporosis detection and monitoring.

## 5. Conclusions

In conclusion, our study unveils the potential of morphometric texture analysis using CT Hounsfield Units in indirectly screening for osteoporosis. The method offers reliable estimates of Bone Mineral Density (BMD) and Bone Mineral Content (BMC), as well as their temporal changes, establishing a novel framework that may enhance osteoporosis management.

## Figures and Tables

**Figure 1 diagnostics-13-03588-f001:**
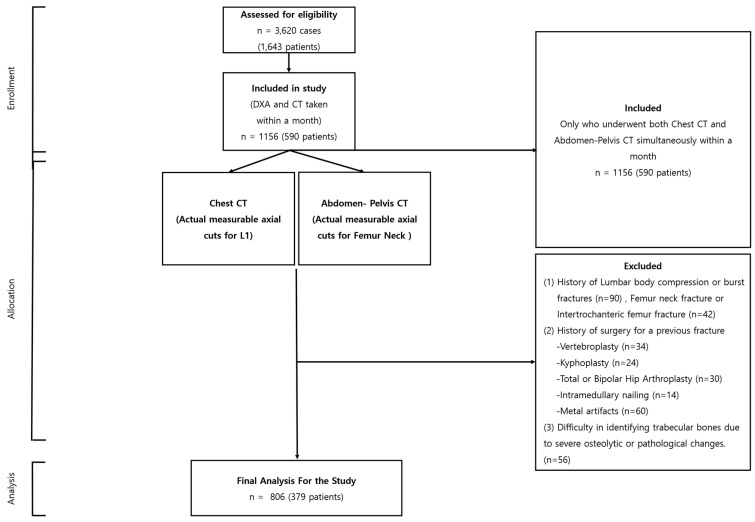
Flowchart depicting the selection of L1 axial cut and femur neck from patients undergoing concurrent chest CT and APCT.

**Figure 2 diagnostics-13-03588-f002:**
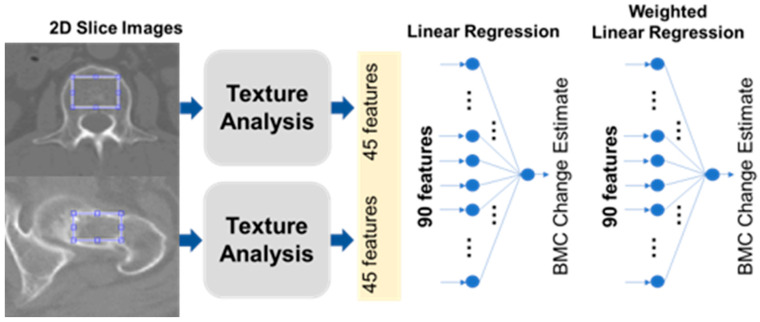
Schematic flow for BMC and BMD estimations from Computed Tomography. BMC, Bone Mineral Content; BMD, Bone Mineral Density.

**Figure 3 diagnostics-13-03588-f003:**
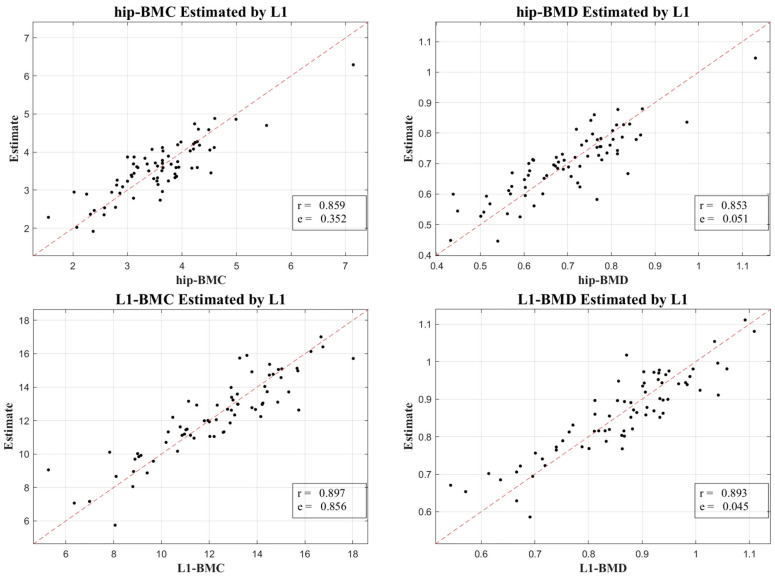
A scatter plot illustrating the correlation between estimated BMC and BMD derived from L1 axial cuts in CT and corresponding measurements from DXA.

**Figure 4 diagnostics-13-03588-f004:**
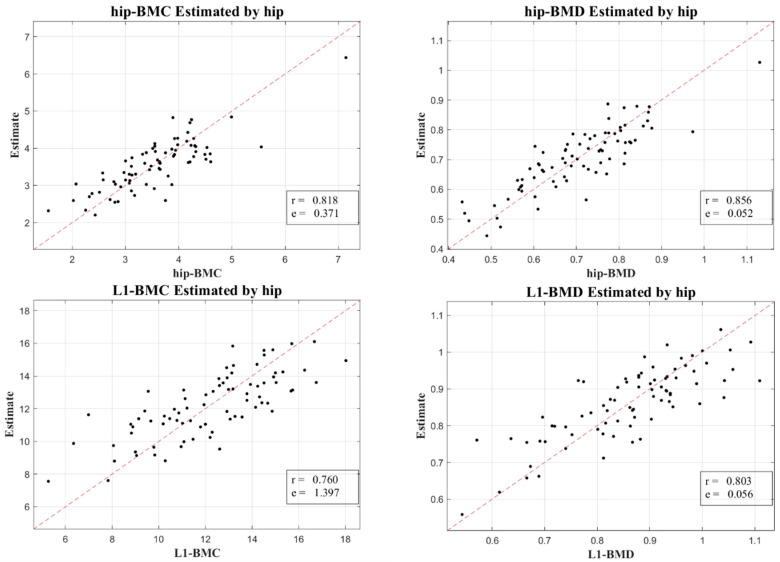
A Scatter plot demonstrating the association between hip-derived BMC and BMD estimations from APCT and corresponding DXA measures.

**Figure 5 diagnostics-13-03588-f005:**
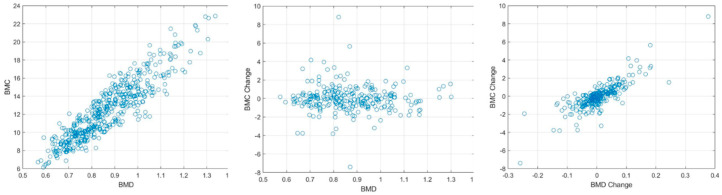
Correlation between BMD and BMC, BMC change, and BMD change. A detailed statistical view of the normalized feature samples utilized in texture analysis.

**Figure 6 diagnostics-13-03588-f006:**
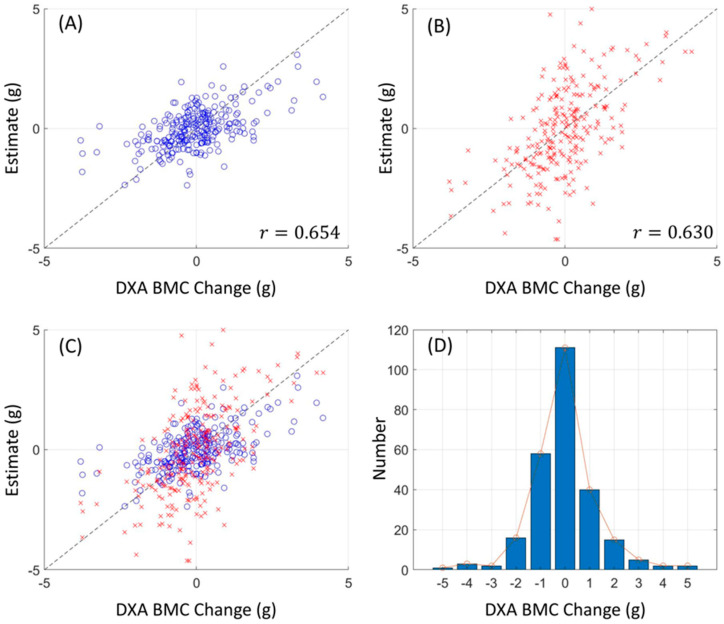
Linear regressor estimation results. The color of each marker represents the value of corresponding patient sample. (**A**,**B**) show the estimates of BMC and BMC change, respectively, over DXA references. The metric ***r*** denotes the correlation coefficient (**C**,**D**).

**Table 2 diagnostics-13-03588-t002:** Demographic data of study participants.

Case (number)	806 (379)
Mean age (years)	54.52 ± 7.56
The time between CT and DXA dates (days)	0.89 ± 5.22
The interval between the first CT and last CT (days)	1412 ± 44.57
The interval between the first DXA and last DXA (days)	1348 ± 31.68
Sex (male/female)	175/204
BMI (kg/m^2^)	23.29 ± 5.65

## Data Availability

The datasets generated and/or analyzed during the current study are not publicly available because we did not obtain authorization from the patients for disclosure regarding patient privacy. However, datasets are available from the corresponding author on reasonable request.

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
