# Peer review of "Comparative Analysis of CT Texture in Lumbar and Femur and Its Correlation with Bone Mineral Density and Content over Time: An Exploratory Study"

_diagnostics, 2023, doi:10.3390/diagnostics13233588_

Round 1
Reviewer 1 Report
Comments and Suggestions for Authors
Dear authors,
Thank you very much for the opportunity to review this paper.
The manuscript is well written, and establishes yet another important and sensible statement regarding the correlation between ct and dxa.
However, there are some minor concerns regarding this paper that i wish to discuss.
How come that appears to be no radiologist among the authors? Involvement of an imaging professional could be beneficial to the manuscript.
I wish that the relative importance of the single feature would be explicited, to grant more reproducible results.
The discussion, and the paper overall, is slightly lacking up-to-date references.
I have recently read a paper on this very topic not long ago on the same journal:
Assessment of Bone Mineral Density from Lumbosacral MRI: A Retrospective Study with Texture Analysis Radiomics
In general, try to compare with the existing literature regarding this topic in the discussion, as it is not scarce.
Otherwise, the methodology is sound and I will be glad to share this with my coworkers if this paper will be published.
Comments on the Quality of English Language
N/a
Author Response
Reviewer 1
Dear authors,
Thank you very much for the opportunity to review this paper.
The manuscript is well written, and establishes yet another important and sensible statement regarding the correlation between ct and dxa.
- Thank you very much for your positive feedback and encouraging comments on our manuscript. We are grateful for your recognition of the manuscript's strengths, particularly in establishing the correlation between CT and DXA. Your insights and constructive feedback are invaluable to us and have greatly assisted in enhancing the quality and clarity of our research.
- We have carefully considered your suggestions and have made the necessary revisions to further improve our manuscript. Please find the detailed responses to your comments in the attached document.
- Thank you once again for your valuable contribution to our work.
However, there are some minor concerns regarding this paper that i wish to discuss.
How come that appears to be no radiologist among the authors? Involvement of an imaging professional could be beneficial to the manuscript.
- Thank you for your insightful comment regarding the composition of our author team. We appreciate your suggestion about the potential benefits of involving an imaging professional, such as a radiologist, in our study.
- We acknowledge that the exclusive participation of orthopedic surgeons in our research team may serve as a limitation to our study. The absence of a radiologist might have led to a certain degree of perspective bias, particularly in the interpretation and analysis of imaging data.
- In addition, we realize that our study might exhibit a possible inclination towards bias, attributed to the predominant reliance on data sourced from a solitary institution. This limitation has been duly noted and will be addressed in the discussion section of our manuscript to provide a more balanced view.
- Thank you once again for your valuable contribution to enhancing the quality of our work.
- We added limitation as follows in Discussion sections.
- “The absence of a radiologist in our team may have limited the scope of our radiological analysis. Future research could be enriched by the inclusion of imaging experts to broaden the interpretive perspectives.”
I wish that the relative importance of the single feature would be explicited, to grant more reproducible results.
- Thank you for your valuable comment regarding the explicit detailing of the relative importance of single features in our study. Your insight is crucial in guiding us to enhance the clarity and reproducibility of our research findings.
- In response to your suggestion, we have revised the manuscript to explicitly address the relative importance of the 45 CT texture analysis features in our linear regression models. These modifications aim to clarify how these features effectively represent the BMC equivalent value and its changes, thereby contributing to the robustness and accuracy of our model.
- We believe that these adjustments will not only address your concerns but also make our results more comprehensible and reproducible for future researchers.
- Thank you once again for your constructive feedback, which has significantly contributed to the improvement of our manuscript.
- We changed our discussion part as follows for the better pointing out the core.
- In summary, our research successfully employs texture analysis of CT Hounsfield Units (HU) to establish linear regression models for estimating Bone Mineral Content (BMC) and monitoring its changes. Among the various factors analyzed, the 45 CT texture analysis features stand out as a key single feature that plays a crucial role in effectively representing the BMC equivalent value and its changes. These features are integral to the model’s ability to mimic the BMC values and track their variations, thereby contributing significantly to the model's robustness and accuracy. The high correlation between these estimates and actual DXA values testifies to the efficacy and precision of our approach. Moreover, the relatively short scan interval between the CT and DXA, compared to previous studies, further reinforces the reliability of our results. These findings lay the groundwork for utilizing CT scans alone in predicting BMC, providing a fresh and efficient perspective in the field of osteoporosis detection and monitoring.
The discussion, and the paper overall, is slightly lacking up-to-date references.
I have recently read a paper on this very topic not long ago on the same journal:
Assessment of Bone Mineral Density from Lumbosacral MRI: A Retrospective Study with Texture Analysis Radiomics
In general, try to compare with the existing literature regarding this topic in the discussion, as it is not scarce.
- Thank you for your insightful observation regarding the need for more up-to-date references in our discussion and throughout the paper. We greatly appreciate your recommendation to incorporate recent literature, particularly the paper titled "Assessment of Bone Mineral Density from Lumbosacral MRI: A Retrospective Study with Texture Analysis Radiomics" published in the same journal.
- In response to your valuable suggestion, we have updated our manuscript to include this reference and have made a thorough comparison with existing literature on this topic in our discussion section. This addition not only strengthens the relevance of our study but also provides a broader context and deeper understanding of the field.
- Your guidance has been instrumental in enhancing the quality and comprehensiveness of our research. We are grateful for your contribution to improving our manuscript.
- The convergence of machine learning with radiomics has witnessed recent endorsements, particularly prevalent in Magnetic Resonance Imaging (MRI) studies (14).
Otherwise, the methodology is sound and I will be glad to share this with my coworkers if this paper will be published.
- We are immensely grateful for your encouraging words regarding the methodology of our study. Your acknowledgment of the soundness of our approach is highly motivating and validating for our research team.
- We are honored by your intention to share our work with your coworkers upon its publication. Such recognition is deeply appreciated and reinforces our commitment to contributing valuable insights to the field.
- Thank you once again for your support and positive feedback, which are integral to the success of our research endeavors.
-----------------------------------------------------------------
Reviewer 2 Report
Comments and Suggestions for Authors
In their manuscript "Comparative Analysis of CT Texture in Lumbar and Femur and its Correlation with Bone Mineral Density and Content Over Time: An Exploratory Study"
the group of authors led by Kim presents a new approach to osteoporosis screening. They use the morphometric texture analysis of CT and compare the results with the values measured in the DXA. Over a period of 8 years, measurements were carried out on over 250 people in over 500 cases. With the help of regression models consisting of 45 attributes, 90 different characteristics could be used to calculate the BMC. The ROI was restricted to the trabecular part of the lumbar vertebral body L1 and the femoral neck. The methodology provides an approach for CT-based osteoporosis screening.
The study does not take into account possible osteoporosis or bone-active drugs, which the authors also cite as a point of criticism. Rather, the authors are interested in showing that CT measurements correlate well with the bone mineral density (BMD) and bone mineral content (BMC) values of a DXA measurement and could supplement or possibly replace these in combination with AI.
My point of criticism: The information on the numbers of examinations and examined persons should be revised. There is a difference between the figures in the abstract and in the manuscript. Different information in the abstract than in the main text. This is not clearly visible, and the reader cannot assign and interpret the corresponding data beyond doubt.
Comments on the Quality of English Language
for an non-native speaker the article is understandable.
Author Response
Reviewer 2
In their manuscript "Comparative Analysis of CT Texture in Lumbar and Femur and its Correlation with Bone Mineral Density and Content Over Time: An Exploratory Study"
the group of authors led by Kim presents a new approach to osteoporosis screening. They use the morphometric texture analysis of CT and compare the results with the values measured in the DXA. Over a period of 8 years, measurements were carried out on over 250 people in over 500 cases. With the help of regression models consisting of 45 attributes, 90 different characteristics could be used to calculate the BMC. The ROI was restricted to the trabecular part of the lumbar vertebral body L1 and the femoral neck. The methodology provides an approach for CT-based osteoporosis screening.
The study does not take into account possible osteoporosis or bone-active drugs, which the authors also cite as a point of criticism. Rather, the authors are interested in showing that CT measurements correlate well with the bone mineral density (BMD) and bone mineral content (BMC) values of a DXA measurement and could supplement or possibly replace these in combination with AI.
- We are immensely grateful for your comprehensive and thoughtful analysis of our manuscript, "Comparative Analysis of CT Texture in Lumbar and Femur and its Correlation with Bone Mineral Density and Content Over Time: An Exploratory Study." Your understanding of our research goals and methodology is deeply appreciated.
- We are particularly thankful for your recognition of our novel approach to osteoporosis screening using CT morphometric texture analysis. Your acknowledgement of the extensive work carried out over 8 years, involving over 250 individuals and 500 cases, is greatly encouraging.
- Regarding your point on the exclusion of osteoporosis or bone-active drugs from the study, we concur that this is a limitation, as you astutely noted. Our primary focus was indeed on demonstrating the correlation between CT measurements and DXA-derived BMD and BMC values, with an eye towards the potential of AI-enhanced screening methods in the future.
- Your insightful comments and the appreciation you have shown for our study not only affirm our efforts but also guide us towards areas for further research and improvement.
- Thank you once again for your invaluable feedback and support. It is greatly motivating and contributes significantly to the advancement of our work.
My point of criticism: The information on the numbers of examinations and examined persons should be revised. There is a difference between the figures in the abstract and in the manuscript. Different information in the abstract than in the main text. This is not clearly visible, and the reader cannot assign and interpret the corresponding data beyond doubt.
- We apologize for any confusion caused by the discrepancies between the numbers of examinations and examined persons reported in the abstract and the main manuscript of our study. We understand the importance of consistency in presenting data and regret any lack of clarity this may have caused.
- In response to your valuable feedback, we have thoroughly reviewed and revised both the abstract and the manuscript to ensure that the information concerning the number of examinations and individuals involved is consistent and accurate. We have made the necessary corrections to eliminate any discrepancies, thereby enhancing the clarity and reliability of the data presented.
- We appreciate your attention to detail and your commitment to maintaining high standards in scientific reporting. Your feedback has been instrumental in improving the quality of our manuscript.
- Thank you once again for bringing this matter to our attention.
- A total of 806 instances (encompassing 379 individuals) were meticulously compiled from a sole institution, during the period stretching from May 6, 2012, to June 30, 2020. In this detailed analysis, each participant was subjected to a pair of chest CT scans, sequentially pursued by a DXA scan, spread over two years. Focused records of BMC values at the inaugural lumbar vertebra (L1) were secured from both the DXA and CT axial slices across all instances.